

# An enhanced approach for automatic annotation of error codes based on Seq2edit

Jian Wang[1,2], Tao Lin[1], Rongsen Zhao[1] and Huiling Zhao[2]

[1] College of Computer Science, Sichuan University, Chengdu, Sichuan, China
[2] College of Electronic Information, Jincheng College, Chengdu, Sichuan, China

## ABSTRACT

The deep natural language translation models have been used for automatic code error correction and have demonstrated outstanding potential. However, a large and accurately annotated training dataset is essential for these models to perform well. The key to improving the performance of these models lies in automatically and accurately annotating code errors and establishing a larger training dataset. Recently, a code error automatic annotation method based on Seq2edit has been proposed to optimize the dataset. However, the accuracy of the annotation is affected because tokens in the input code from the same statement may be aligned to different statements. This article proposes a Seq2edit annotation method based on the source code's sentence structure. By dividing the code into statements with independent meanings and introducing a cost coefficient to improve the Levenshtein algorithm, this method optimizes the calculation of edit distance and enhances the ability to align tokens. Experimental results show that this method can fully utilize the contextual information of the source code during the automatic annotation process, leading to a significant improvement in annotation accuracy.

# INTRODUCTION

## Motivation

With the extensive unsupervised pretraining of deep neural network models on a large amount of source code, the ability of machines to understand code has improved significantly (*Ahmad et al., 2021*). Impressive performance has been achieved in tasks such as code classification, code querying, code completion, and documentation generation (*Lu et al., 2021*). However, code correction, which requires strong reasoning, still faces significant challenges (*Allamanis, Jackson-Flux & Brockschmidt, 2021*). Code correction is the neural machine translation task (NMT) that translates an erroneous task into a correct task. Each word in the code is tokenized as a token, and the code sequence containing errors is directly translated into the target token sequence through a Seq2seq approach, thereby obtaining the output target code (*Malmi et al., 2019*). This end-to-end approach is simple and fast, but it suffers from low efficiency due to the need to predict all tokens, leading to lower similarity between the output and input code. Subsequently,

Corresponding author
Tao Lin, lintao@scu.edu.cn

*Malmi et al. (2019)* and others proposed the "Seq2edit" method to improve this situation. Instead of generating a target token sequence, it outputs an edit operation sequence for the input code's token sequence. Each input token is annotated with a category label, such as "KEEP", "DELETE" or "INSERT" indicating which tokens in the input should be retained, deleted, or have new tokens inserted. During the generation of the target token sequence, the model can directly perform stretching and deletion operations on the positions of tokens based on the labeled annotations. After training, the deep learning model only needs to predict the content for "INSERT" type labels (*Yao et al., 2021*; *Berabi et al., 2021*). This approach significantly reduces the number of tokens that need to be predicted and leverages the kept context in the target code to improve the accuracy of predicting the missing words (*Zhu et al., 2021*; *Chen et al., 2019*). Therefore, the accuracy of token annotations in the Seq2edit approach is crucial for generating the final token sequence. Establishing a larger training dataset to improve the model's prediction label accuracy is the key to its success.

The word-based annotation method used in the Seq2edit approach was the earliest one adopted (*Malmi et al., 2019*). It involved labeling each token in the input with an edit-type label to indicate the differences between the input and output sequences, which were determined based on the longest common subsequence matching. Although this method was simple, most input tokens were annotated with the "KEEP" label, resulting in highly imbalanced label types in the training samples. This imbalance led to the model being biased towards predicting unknown tokens as the "KEEP" type. A method of span merging for edit sequences was proposed to address the problem of label type imbalance which may mislead the model during training (*Stahlberg & Kumar, 2020*; *Hu et al., 2022*). The term "span" refers to a region range of label positions where consecutive identical labels can span across multiple lines of code statements for merging, as long as they adhere to the principle of minimum edit distance. This process generates the target code sequence, resulting in a collection of triples in the form of (Tags, and Span Token), which effectively mitigates the imbalance in types. The minimum edit distance employed during merging refers to the fewest steps, such as adding, deleting, or replacing label pairs, required to transform the input code into the correct output code. One typical edit distance algorithm is Levenshtein, which uses dynamic programming and backtracking to find the path with the minimum total edit distance (*Wang et al., 2017*). The adjacent operations of the same type along the edit path were combined to obtain the output annotation sequence. However, a significant limitation of such methods is that multiple different edit paths often exist when determining the minimum edit distance. The deterministic backtracking algorithm usually selects only one path that aligns all words forward or backward. As shown in the correction sequence 1 in Table 1. The annotations lead to situations where multiple words that should belong to a single code statement are segmented and aligned to different statements. In contrast, correction sequence 2 represents a more concise and clear expected annotation result.

Therefore, to address the issues of imbalanced label categories caused by word-based methods and errors in training data annotation due to dispersed alignment of tokens in span-based methods, this article proposes a sentence label method (SLM) for code

**Table 1 Comparison of different annotation methods for code errors.**

| | |
|---|---|
| Error code | public java.util.List <TYPE_1> METHOD_1 ( ) { **for ( TYPE_2 VAR_2 : VAR_3 ) {VAR_1 . METHOD_2 ( VAR_2 . METHOD_1 ( ) ) : }** return VAR_1 ; } |
| Correct code | public java.util.List <TYPE_1> METHOD_1 ( ) { return VAR_1 ; } |
| Error-corrected sequence 1 | (KEEP, 6, NONE), **(DELETE, 10, NONE)**, (KEEP, 11, NONE), **(DELETE, 15, NONE)**, (KEEP, 17, NONE), **(DELETE, 29, NONE)**, (KEEP, 33, NONE) |
| Error-corrected sequence 2 | (KEEP, 9, NONE), **(DELETE, 29, NONE)**, (KEEP, 33, NONE) |

statement alignment, aiming to enhance the accuracy of automatic data annotation for training corpora. This method employs a strategy of aligning and merging code statements, enhancing the Levenshtein algorithm and beam search approach, increasing the probability of selecting correct tokens from the input code sequence and striving to retain the original form of the input code as much as possible. This contributes to establishing a larger training dataset and enhances the effectiveness of training complex code correction models.

## RELATED WORK

Currently, code correction work can be mainly divided into two categories. The first category is based on Seq2seq end-to-end approaches, where large amounts of data are used to pre-train the model. In this approach, the erroneous source code is encoded and used as input to generate the corrected output code through an autoregressive process. Although these methods achieved good correction results, the generated code had a significant edit distance from the erroneous code. The second category is based on the Seq2edit approach, which involves a pipeline for code correction. In this method, the erroneous source code is first labeled with edit categories such as "KEEP", "INSERT" and "DELETE" and then the erroneous source code is operated upon based on these labels. This approach tends to preserve the content of the input source code well, but the correction performance is not entirely satisfactory.

### A Seq2seq-based end-to-end neural network error correction approach

End-to-end neural network models typically adopt the Transformer architecture. They are pre-trained on a large *corpus* of data and then use a decoder to generate the target sequence directly. *Kanade et al. (2020)* first proposed using the CuBERT model based on the BERT model for code pre-training. Their article used Python code from the Google BigQuery platform as the *corpus* to learn contextual embeddings for code. They successfully achieved variable misuse detection, binary operator error detection, and operator swapping for error correction. *Feng et al. (2020)* also proposed the CodeBERT model, which utilizes a dual-modal pre-training approach for both code and natural language to learn more general token representations. Following this, *Mashhadi & Hemmati (2021)* attempted to fine-tune CodeBERT on the ManySStuBs4J dataset to repair simple JAVA code errors.
*Zhu et al. (2021)* introduced the GraphCodeBERT model, which leverages data flow to represent the source and destination of variable data more effectively. They performed pre-training on code, comments, and variable sequences to repair C language code and achieved high repair accuracy, though it was limited to variable type errors. The articles mentioned above mainly used encoder architectures, while some researchers focused on using decoder architectures for code correction. *Svyatkovskiy et al. (2020)* proposed the GPT-C model, which applied the generative pre-trained Tranformer (GPT) architecture for code correction attempts. This model was primarily used for code completion tasks, and they proposed four training strategies. However, its performance in multilingual settings was not satisfactory.

## Code correction method based on seq2edit code sequence labeling

The Seq2seq approach generates the target code from scratch, which leads to low efficiency in target code generation due to the need to select words (tokens) from a huge vocabulary. Some Seq2edit methods have been proposed to address this issue. *Gu et al. (2016)* introduced a generation mechanism allowing selective copying from the input text. However, this copying mechanism still requires generating words not present in the input text, which does not fundamentally solve the problem of low efficiency caused by a large vocabulary. To reduce the vocabulary size, *Malmi et al. (2019)* proposed a method for generating the vocabulary by aligning erroneous code and correct code based on their longest common subsequence during annotation. As shown in Table 2, the "public" in the beginning is labeled with a "KEEP" tag, indicating that it should be retained.

The remaining parts are first labeled with a combination of "DELETE" and "KEEP$^{word}$" indicating that the unnecessary tokens in the erroneous code should be deleted, and then the correct content is added sequentially. For example, in Table 2, if the erroneous code "void" should be replaced with "boolean," it would be represented as "DELETE KEEP$^{boolean}$." This approach significantly reduces the vocabulary size, consisting of only one "DELETE" label and multiple "DELETE+KEEP$^{word}$" labels. Only the most probable vocabulary is selected from the vocabulary table during prediction.

*Omelianchuk et al. (2020)* improved this method by introducing the "INSERT$^{word}$" label to replace the original "KEEP$^{word}$" and "DELETE$^{word}$" labels. They also added the "REPLACE$^{word}$" label to represent replacement operations. When comparing different parts of the erroneous and correct code, not all words are included in the vocabulary table; only frequently occurring words are selected to be added. However, this may lead to Out-Of-Vocabulary (OOV) situations where some words are not present in the vocabulary.

*Stahlberg & Kumar (2020)* further improved the method by proposing a span-based annotation approach. This method also uses algorithms like the longest common subsequence and Levenshtein distance (*Levenshtein, 1966*) to align erroneous and correct code, and it adopts "KEEP", "DELETE", "INSERT" and "REPLACE" tags as edit categories. The main difference is that the tags are independent and not combined with words. After obtaining the correct sequence of edit operations, the same tags can be merged into triplets representing edit operations (Tags, Span, Token). These triplets indicate the edit operation category, the ending position of the span, and the predicted result, respectively. As shown

**Table 2  Word-based tagging methods.**

| Error code | *public* **void** *METHOD_1 (TYPE_1 node) { METHOD_2 (node) ; METHOD_3 (VAR_1 . get(((VAR_1 . size()) −1))) }* |
| --- | --- |
| Correct code | *public* **boolean** *METHOD_1 (TYPE_1 node) {* **boolean set** *= METHOD_2 (node) ; if (set) METHOD_3 (VAR_1 . size()) −1))) ;* **return set ;** *}* |
| Error-corrected sequence | KEEP DELETE **KEEP**[boolean] KEEP KEEP KEEP KEEP KEEP KEEP **KEEP**[boolean set=] KEEP KEEP KEEP KEEP **KEEP**[if (set)] KEEP KEEP KEEP KEEP KEEP KEEP KEEP KEEP KEEP KEEP KEEP **KEEP**[return set;] |

**Table 3  Span-based tagging methods.**

| Error code | *public* **void** *METHOD_1 (TYPE_1 node) {*<br>*METHOD_2 (node) ;*<br>*METHOD_3 (VAR_1 . get(((VAR_1 . size()) −1))) ;*<br>*}* |
| --- | --- |
| Correct code | *public* **boolean** *METHOD_1 (TYPE_1 node) {*<br>**boolean set** *= METHOD_2 (node) ;*<br>**if (set)** *METHOD_3 (VAR_1 . get(((VAR_1 . size()) −1))) ;*<br>**return set ;**<br>*}* |
| Error-corrected sequence | (KEEP, 1, NONE), **(REPLACE, 2, boolean)**, (KEEP, 8, NONE),<br>**(INSERT, 8, boolean set =)**, (KEEP, 13, NONE), **(INSERT, 13, if (set))**, (KEEP, 33, NONE),<br>**(INSERT, 33, return set ;)**, (KEEP, 34, NONE) |

in Table 3, the triplet (REPLACE, 2, boolean) indicates that all the words between the ending position of the previous span and the second position need to be replaced with "boolean."

Table 3 shows that this method only needs to output one span, represented by a triplet, to encompass all the operations within that span. For the spans labeled as "INSERT" and "REPLACE," the prediction model generates the required words. This approach efficiently handles spans labeled as "KEEP" and "DELETE" without the need for further operation predictions. Therefore, it is much more efficient than word-based annotation methods and effectively solves the problem of out-of-vocabulary (OOV) words.

# MATERIALS AND METHODS

## SLM method: concept and implementation

The SLM is an improved algorithm based on the seq2edit span-based approach. The positional order of statements in code does not uniformly affect the outcome. For instance, some statements, such as variable assignment statements, may have no impact on the result regardless of their position before usage. In contrast, the position of other statements, such as variable definitions, can significantly alter the outcome if swapped with adjacent statements. However, current span-based methods, which rely on the longest common subsequence and Levenshtein distance algorithms, may result in unreasonable alignment positions during merging. This can be addressed by improving the token

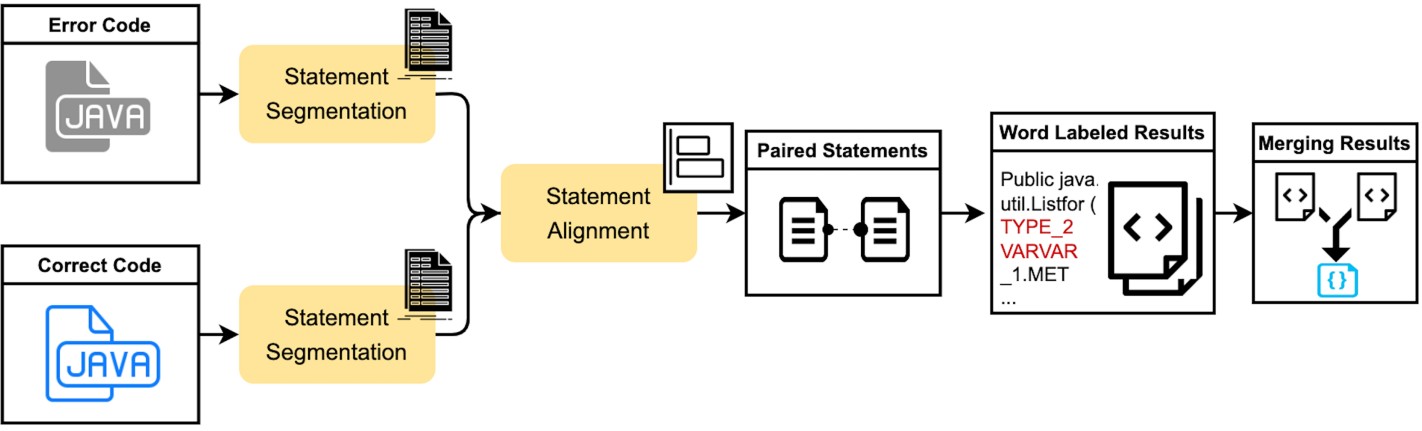

**Figure 1** The schematic diagram of the process of the SLM method.               

alignment approach. The proposed SLM method segments the code into statements and employs an enhanced Levenshtein distance algorithm to align code statements and tokens. By incorporating a cost function into the original computation, the method mitigates the interference caused by differences in variable or function names during edit distance calculation. This avoids redundant or erroneous triplet results due to merging, thereby yielding a more optimal span annotation outcome. The main workflow of this method is illustrated in Fig. 1 and consists of three critical modules: source code statement segmentation, statement alignment computation, and result span annotation.

## Statement segmentation

Statement segmentation is similar to Chinese word splitting, where a piece of code is divided into statements as the smallest units. In JAVA programs, code statements are typically categorized into basic and compound statements. Basic statements are separated by semicolons, front curly braces, and back curly braces, such as assignment statements with a semicolon at the end. Compound statements consist of multiple basic statements, such as functions, loops, and conditional statements. The opening curly brace "{" is treated as an end symbol, and the closing curly brace "}" is merged into the end of the preceding statement as a basic statement. Taking a function as an example, the code can be divided into statement blocks as as shown in Table 4.

Take a pair of erroneous and correct code data, denoted as <X, Y>, and segment them into statements as $S(X) = (X_1..X_n)$ where $X_i$ represents a set of statements obtained by segmenting the code. Similarly, following the same approach, we get $S(Y) = (Y_1..Y_m)$, where each $Y_i$ represents a set of statements obtained by segmenting the correct code.

## Statement alignment calculation

After segmenting the correct and erroneous statements, we employed candidate-aligned statements and a beam search algorithm to identify the optimal matching solution. For the candidate-aligned statements, we utilized an optimized adjast_Levenshtein algorithm to compute the optimal pairing of statements. The relationships between the algorithms are shown in the Fig. 2.

**Table 4 The result of code segmentation.**

| | |
|---|---|
| Error code $X_1$ | Public java.util.List <TYPE_1> METHOD_1 () { for (TYPE_2 VAR_2: VAR_3) { VAR_1.METHOD_2 (VAR_2.METHOD_1 ( )); }} return VAR_1 ; } |
| Statement 1 {$X_1$} | Public java.util.List <TYPE_1> METHOD_1 () { |
| Statement 2 {$X_2$} | for (TYPE_2 VAR_2: VAR_3) { |
| Statement 3 {$X_3$} | VAR_1.METHOD_2 (VAR_2.METHOD_1 ( )); } |
| Statement 4 {$X_4$} | return VAR_1 ; } |

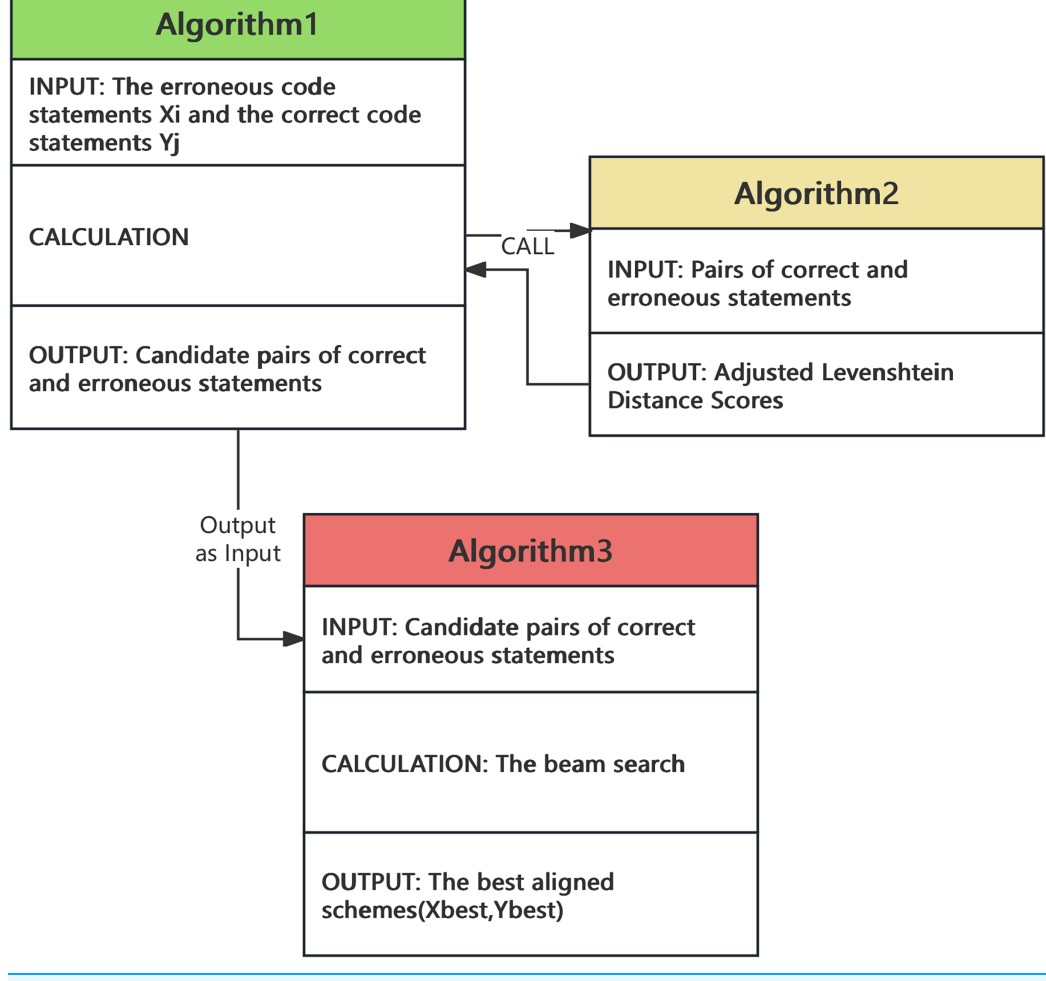

**Figure 2 The relationships between the algorithms.**

The divided erroneous code statements ($X_1..X_n$) and correct code statements ($Y_1..Y_m$) are aligned pairwise. Each element in the $X$ set is compared with each element in $Y$ set to calculate their similarity scores using the adjast_Levenshtein. The top $K$ pairs of statement

---

**Algorithm 1** Candidate combination algorithm.

**Require:** *Segmented statements of error code $X = \{X_1, X_2, \ldots, X_n\}$;*
  *Segmented statements of correct code $Y = \{Y_1, Y_2, \ldots, Y_m\}$*
**Ensure:** The combination of candidate alignments for error codes and correct codes
  $O = \{O_1, O_2, \ldots, O_{k*n}\}$
  1: $O \leftarrow []$; $T \leftarrow []$
  2: **for** $i = 0 \rightarrow n - 1$ **do**
  3:   **for** $j = 0 \rightarrow m - 1$ **do**
  4:     $T.append(getCost(X_i, Y_j))$ #Calculate the edit distance between $X_i$ and $Y_j$.
  5:   **end for**
  6:   #Take the k groups with the smallest edit distance in $T$
  7:   #and add them to the alignment combination $O$.
  8:   $O.append(getMinGroup(T, k))$
  9:   $T.clear()$
 10: **end for**
 11: **return** $O$

---

blocks with the highest scores are added to the span candidate set $O$. As a result, $O$ gets a total of $n * K$ candidate pairs $((X_1, Y_1) \ldots (X_i, Y_k))$. The edit distance algorithm is used for similarity calculation, described in Algorithm 1.

In computing the edit distance between a pair of statements, the conventional Levenshtein algorithm is limited by its reliance on character-level insertions, deletions, and substitutions. It fails to account for word-order transpositions due to its inability to capture character sequence changes. Therefore, we adjust the word order to align the sequences properly, and then we apply the adjust_Levenshtein to calculate the distance. In this context, we defined the $getCost(X_i, Y_j)$ function in the Algorithm 1 to compute the edit distance between statements of $X_i$ and $Y_j$ dynamically. The formula employed in the function is defined in Eq. (1).

$$adjust\_Levenshtein(X_i, Y_j) = Levenshtein(X_i, Y_j) \times Cost(X_i, Y_j). \tag{1}$$

To detect and correct word-order discrepancies, we segment statements $X_i$ and $Y_j$ into individual words $x_i$ and $y_j$ using spaces as delimiters. If the words extracted from two statements can be reordered to form identical sentence structures, this is identified as a word-order error. When such errors are present, minimal editing operations are applied to identify feasible word-order adjustments between the two statements and compute the associated cost. Here, We use the cost function $Cost(X_i, Y_j)$ to represent the word order swapping distance, which helps us choose the pair $(X_i, Y_j)$ with the lowest cost.

For the alignment of $X_i$ and $Y_j$, we need to consider both the category and similarity factors. Firstly, we use regular expressions and heuristic rules to determine the category $Type(X_i, Y_j)$ belongs to, such as assignment statements, loop statements, and function call statements. If the categories are equal, then $Cost_{type}$ is set to 1; otherwise, it is set to $\lambda$. We set the penalty value $\lambda = 3$ to increase the penalty so that statements of the same functional

---

**Algorithm 2**  **The adjust_Levenshtein algorithm (getCost($X_i, Y_j$) function) for word order permutation.**

**Require:** The basic statements $X_i$ and $Y_j$ to be aligned.

**Ensure:** Flag indicating whether it is a program error problem; $lev_{trans}$: Levenshtein edit distance for swapping.

1: $L_x = \text{Len}(X_i); L_y = \text{Len}(Y_j)$
2: $X = \text{lowercase}(X_i); Y = \text{lowercase}(Y_j)$
3: $flag = \text{False}; Lev_{trans} = Lev_{L_x L_y}$
4: **for** $k = 1 \rightarrow \min(L_x, L_y)$ **do**
5:    **if** $\text{sorted}(X[L_x - k : L_x]) = \text{sorted}(Y[L_y - k : L_y])$ **then**
6:       $flag = \text{True}$
7:       $Lev_{trans} = \min(Lev_{L_x L_y}, switch\_counts(L_k) + Lev_{L_x - k L_y - k})$
8:    **end if**
9: **end for**
10: $Lev_{trans} = Lev_{trans} * Cost(X_i, Y_j)$
11: **return** $flag$, $Lev_{trans}$

---

category are prioritized for alignment. Furthermore, we use a lexical analyzer to extract all identifiers involved in $X_i$ and $Y_j$, such as method names, function names, and variable names. The overlap $\text{Cost}_{name}$ between those sets of identifiers is then calculated. The overlap is defined as the proportion of common elements shared between the strings (*e.g.*, method names, function names, and variable names). If the overlap is identical and all elements are shared between them, meaning two sentences are identical up to swaps, the $\text{Cost}_{name}$ is set to 1; otherwise, it is set to 0. The calculation is shown in Eq. (2).

$$Cost(X_i, Y_j) = Cost_{type} - Cost_{name} \tag{2}$$

The implementation of the edit distance algorithm incorporating the cost function, $getCost(X_i, Y_j)$, is detailed in Algorithm 2. The algorithm takes the basic statements $X_i$ and $Y_j$ that require alignment as input. After computation using the adjust_Levenshtein, Algorithm 2 outputs a flag indicating whether a word-order transposition has occurred and the final edit distance score $Lev_{trans}$. The flag is utilized to facilitate the analysis of the number of statements involving word-order transpositions, while the Levtrans score is employed to aid in the selection of candidate pairs in Algorithm 1.

To identify the alignment scheme with the minimal cost among all possible alignments, we first select the K pairs with the smallest edit distances from the set of alignment candidates $O$ output by Algorithm 1. These pairs are used to initialize K alignment schemes, where each scheme $P_j$ contains a pair of aligned code statements. These schemes are stored in the set $P = \{P_1, P_2, \ldots, P_k\}$. Meanwhile, for each scheme, we maintain a set $U = \{U_1, U_2, \ldots, U_k\}$ of already aligned statements to prevent the repeated selection of the same statements in iterations. Subsequently, in each round, we continue to select K pairs from $O$ for alignment operations, ordered by increasing edit distance. If no alignable pairs are available for a particular scheme, a basic statement is randomly selected and added to the

alignment scheme. This implies that the remaining basic statements need to be inserted or deleted, with their edit distances being equivalent to their string lengths. Consequently, each existing alignment scheme $P_i$ generates up to K new branching schemes, resulting in $K^2$ new schemes in total. We employ a beam search approach, retaining only the K schemes with the lowest cumulative edit distances after each round, until all basic statements are aligned. Ultimately, the scheme with the minimal total edit distance is selected as the optimal alignment scheme S from all candidates. This process is illustrated in Algorithm 3, where we utilize the Top-K alignment algorithm based on beam search to obtain the optimal alignment scheme after obtaining the candidate pairs from Algorithm 1.

## Span annotation

After obtaining the S set through the abovementioned calculations, the original span annotation method is applied to label each word accordingly before merging, resulting in a collection of triplets. As a result, given a pair of erroneous and correct code samples as input, an edit sequence output can be generated, where each element in the sequence consists of a triplet.

## Selection for K-value

The choice of K-value in Algorithms 1 and 3 is vital. When K is too small, the alignment algorithm may overlook high-quality alignment combinations during the selection process. Conversely, when K is excessively large, the algorithm may select numerous alignment combinations that offer little improvement to the overall alignment quality. This not only fails to enhance the quality of the alignment results effectively but also significantly increases the computational time required for the algorithm to execute. To address this issue, we conducted a series of parameter-tuning experiments to determine the optimal value of K for the beam search width in our alignment algorithm. Specifically, we performed alignment tests on a manually annotated dataset and compared the alignment outcomes for different values of K. The results of these experiments are illustrated in Fig. 3.

The results of the K-value selection experiment reveal several critical insights. As the value of K increases, there is a significant improvement in the annotation accuracy of the training set by the annotation algorithm. Furthermore, when the annotated results are utilized to train a code correction model, the accuracy of the generated code by the model also increases with the value of K. For instance, compared to K = 2, when K = 5, the annotation accuracy (ACC) increased by 28.0%. However, the time required to annotate the same training set increased by approximately 107%, with the computational time being proportional to the value of K. Notably, when K was set to 6, there was no significant change in the ACC or the interval number metrics compared to K = 5, while the execution time of the algorithm increased dramatically. Therefore, considering the trade-off between annotation accuracy and computational efficiency, the optimal value of K was determined to be 5.

---

**Algorithm 3** The alignment calculation for TopK.

**Require:** The combination of candidate alignments for error codes and correct codes: $O = \{O_1, O_2, \ldots, O_{5n}\}$;

**Require:** Number of alignment schemes retained: $k$

**Ensure:** Alignment schemes: $S$

1: $P = [P_1, P_2, \ldots, P_k]$ {Initialize $k$ empty candidate alignment schemes.}
2: $U = [U_1, U_2, \ldots, U_k]$ {Initialize the set of already aligned code statements in each alignment scheme.}
3: $P_{\text{selected}} = []$
4: Sort the aligned candidate combinations in ascending order of edit distance.
5: Sort($O$)
6: **for** $i = 0$ to $n - 1$ **do**
7:     **for** $j = 0$ to $k - 1$ **do**
8:         #Select $k$ combinations with the smallest cost from the non-intersecting set between aligned candidate combinations $O$ and $U_j$.
9:         $O_{\text{selected}} = \text{getTopKButNotUsed}(O, j)$
10:         #If the combination is not empty, then up to $K$ branching combinations are generated.
11:         #for each existing scheme
12:         #If the combination is empty, keep the current scheme.
13:         **if** $O_{\text{selected}} \neq \text{null}$ **then**
14:             $P_{\text{selected}}.\text{append}(\text{concat}(P_j, O_{\text{selected}}))$
15:         **else**
16:             $P_{\text{selected}}.\text{append}(P_j)$
17:         **end if**
18:     **end for**
19:     #Keep $k$ schemes with the smallest edit distance and generate the set of aligned statements corresponding to each scheme.
20:     $P = \text{getTopK}(P_{\text{selected}})$
21:     $U = \text{getOccupyFromPlan}(P)$
22:     $P_{\text{selected}}.\text{clear}()$
23: **end for**
24: $S = \text{getMin}(P)$
25: **return** $S$

---

# EXPERIMENT

The experimental process consisted of two main experiments:

- Annotation accuracy evaluation experiment: The first experiment compared the performance of automated code sequence labeling on the manually labeled dataset to assess the effectiveness of the proposed method.
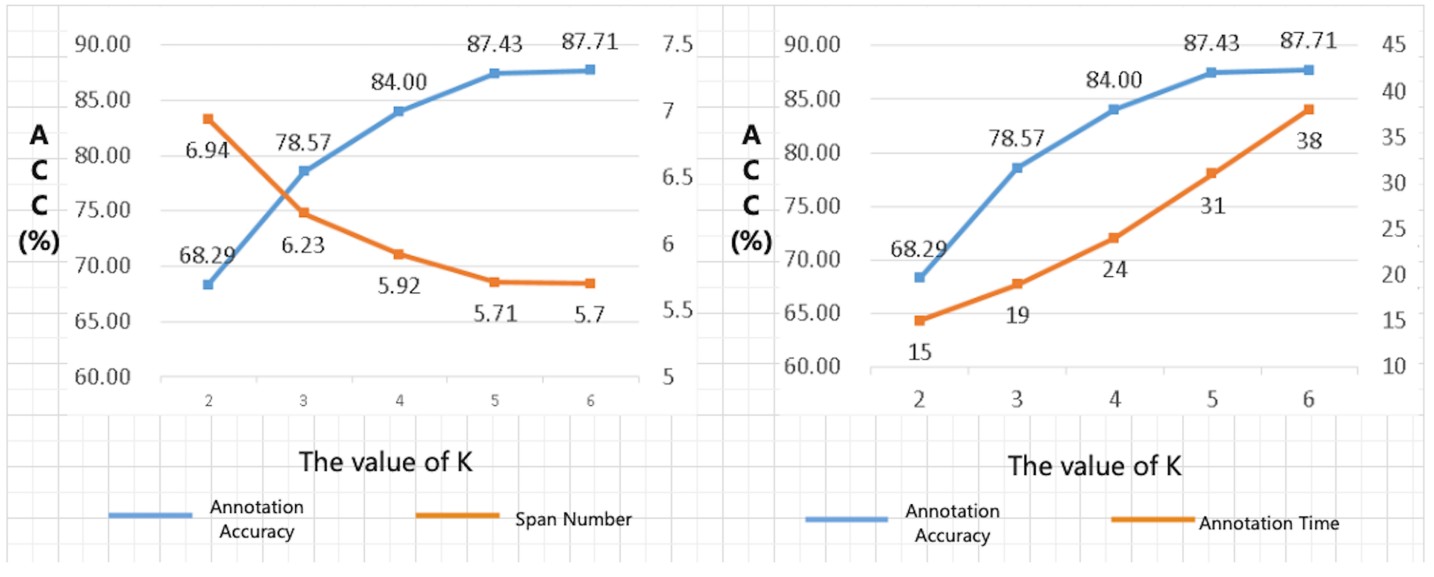

**Figure 3  Results of K-value selection experiment.**

- Model effectiveness validation experiment: The second experiment involved annotating the entire CodeXGLUE Small dataset using our method and two existing annotation methods. We trained models using the annotated results and assessed the impact of the corpora on the models by comparing the corrected code with the correct code.

In summary, the experiments aimed to validate the effectiveness of the method through code sequence automatic labeling on the manually labeled dataset and assess the influence of different labeling results on model performance through training and evaluation.

## Datasets

To assess whether our annotation method brings improvements to code correction models, we conducted experiments using the CodeXGLUE Small dataset and manually labeled datasets (*Lu et al., 2021*). The CodeXGLUE Small dataset consists of 46,680 pairs for the test set, 5,835 pairs for the validation set, and 5,835 pairs for the test set, with each code pair containing a segment of erroneous code and its corresponding correct code. The manually labeled dataset was randomly selected from the CodeXGLUE Small dataset, comprising 350 code pairs. The annotation principle was to correct the erroneous code to the correct code using the minimum number of editing operations while preserving the overall structure of the code. The third-party dataset is available at: https://github.com/microsoft/CodeXGLUE.

## Data preprocessing

We performed preliminary data preprocessing on the CodeXGLUE Small dataset, which consists of code text. First, we used the data cleaning tool *Numpy* to remove noise and irrelevant information from the code. Specifically, we applied *np.char.strip* to remove

leading and trailing whitespace from each code segment and *np.char.replace* to eliminate unnecessary whitespace within the code lines. Additionally, we removed comments using regular expressions to focus solely on the executable code. Second, we identified and removed outliers in code length using the *zscore* method from the statistical package *SciPy*, ensuring that the dataset contained only code segments within a reasonable length range. Finally, we eliminated duplicate code segments using the *drop_duplicates* function from the data deduplication tool *Pandas*. The preprocessed datasets are archived and readily accessible at the following repository: https://github.com/Ling0924/codebert_gec//tree/main/data. Furthermore, the data generated as a result of our analyses is deposited in a distinct section of the same repository, which can be accessed *via* this link: https://github.com/Ling0924/codebert_gec/tree/main/data/modelinput.

## Model

We chose CodeBERT for this study for several reasons. CodeBERT is trained on large code datasets, making it well-suited for programming tasks. It also outperforms models without code-specific pre-training in benchmark tasks and its transformer architecture effectively handles the complexities of source code. Finally, strong community support ensures CodeBERT stays up-to-date with the latest advancements. In our project, we have elected to pre-train CodeBERT utilizing a seq2seq methodology augmented with cross-attention. This approach is designed to enhance the model's proficiency in comprehending and generating code, thereby equipping it to navigate the nuanced patterns and interdependencies inherent to coding tasks with greater finesse.

## Computing infrastructure

In the context of our research, both macOS and Windows operating systems have been identified as suitable platforms, with no particular hardware prerequisites stipulated. Participants are instructed to engage with the README file, which is optimally accessed and manipulated within the confines of a Jupyter Notebook environment (.ipynb file). This approach ensures a standardized and efficient interaction with the provided documentation and associated computational resources.

## Labeling methods and prediction models

In the experiment comparing prediction results using the CodeBERT model, we selected the word-based annotation algorithm proposed by *Malmi et al. (2019)* and the span-based annotation method proposed by *Stahlberg & Kumar (2020)* as benchmarks for comparison. The trained model selected for this study was CodeBERT (*Feng et al., 2020*) which serves as the baseline model for code repair tasks in the CodeXGLUE (*Lu et al., 2021*) framework.

## Evaluation metrics

The evaluation metrics include accuracy (ACC), bilingual evaluation understudy (BLEU), edit spans, and average processing time. ACC represents the proportion of generated annotation sequences that match the manually annotated results exactly.

BLEU (*Papineni et al., 2002*) is a widely used metric that evaluates the quality of machine translation introduced by researchers, including *Papineni et al. (2002)*. It aims to address the time-consuming and costly nature of traditional human evaluation methods. BLEU primarily calculates the similarity between translation results and reference answers based on n-gram matching and sentence length penalty. It helps developers quickly and accurately assess the performance of machine translation systems. BLEU values are higher when the source code sequences and result sequences are closer. The metric "number of edit spans" denotes the average count of triplets representing edit operations within the results, thereby reflecting the number of edit operations performed. Due to differences from other labeling methods, the word-based labeling method converts its labeling results into a similar span representation format to the span-based labeling method, following the principle of "merge if adjacent and identical." For unused "REPLACE" edit labels, a method is adopted to merge adjacent "INSERT" and "DELETE" labels into "REPLACE" for a fair comparison.

Average processing time refers to the average time it takes for the annotation sequence results to be obtained through the SLM method after processing the input code. It is calculated by dividing the total time spent on all code annotations by the number of code samples. These evaluation metrics provide a comprehensive assessment of the performance and efficiency of the proposed method.

## RESULTS AND ANALYSIS

### Annotation accuracy evaluation experiment

In the annotation accuracy evaluation experiment, the search parameter K was set to 5, and a comparative study was conducted between the SLM method and the word-based and span-based labeling methods on the manually labeled dataset. As shown in Table 5, the experimental results present the accuracy, number of edit spans, and processing time for the three labeling methods.

The experimental results reveal that the SLM-based labeling method significantly improved training set annotation accuracy, reaching the optimal value of 87.43%, without a substantial increase in edit span count compared to the other two methods. The improvement percentages were 44.35% and 22.90%, respectively, indicating that the statement-based labeling approach is closer to real human annotations. Meanwhile, the edit span count was 5.71, indicating an average of 5.71 triplets per generated edit sequence, falling between 6.83 and 4.97 for the other two methods. This indicates the proposed method selectively merges spans to preserve the code statement structure. The average annotation time was recorded as 31 s, significantly higher than the other two methods, signifying that this method requires more time.

The experiment demonstrates the effectiveness of the SLM-based labeling method in enhancing annotation accuracy while maintaining a reasonable edit span count, showcasing its potential for more accurate and context-aware code error annotation.

Table 5 **The comparison of annotation methods and annotation effectiveness.** The bold entries represent the optimal results in the comparison.

| Methods | ACC (%) | Edit span count | Time (s) |
| --- | --- | --- | --- |
| Word_based | 60.57 | 6.83 | **10** |
| span_based | 71.14 | **4.97** | 15 |
| Our method (K = 5) | **87.43** | 5.7 | 31 |

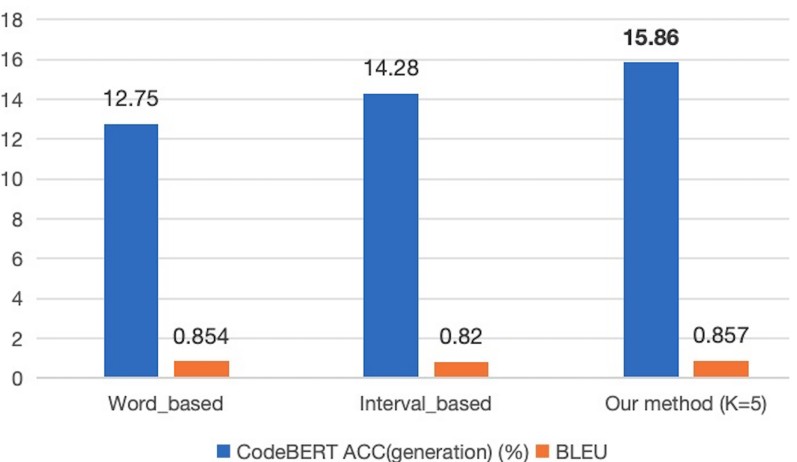

Figure 4 **The comparison of prediction performance of different models under various annotation methods.**

## Model effectiveness validation experiment

To further assess the impact of the SLM annotation method on the performance of predictive models, we indirectly evaluated the effectiveness of our annotation approach by measuring the code generation accuracy of the models. Given that the CodeBERT model employs bidirectional feature extraction, which enhances its performance in context-dependent tasks such as cloze tests, it is particularly well-suited for our experimental tasks. Therefore, in the second experiment, we trained the CodeBERT model on the entire CodeXGLUE Small dataset using three different annotation methods and compared the code correction results on the test set. The experimental results are presented in Fig. 4.

The results of this experiment provide insights into how the SLM model-annotated sequence samples contribute to the performance of the CodeBERT model in the context of the error correction task. The comparison sheds light on the effectiveness of utilizing SLM-generated annotations for improving the performance of prediction models.

The experimental results demonstrate that in the CodeBERT model, the proposed SLM labeling method achieved the best accuracy and BLEU performance. The accuracy value of 15.86% is an improvement of 3.11% compared to the word-based method and 1.58% compared to the span-based method. This suggests that the *corpus* labeled using this method enhances prediction performance after model training. Furthermore, the BLEU

**Table 6  The comparison of test sample outputs using CodeBert.**

| Input code | public void method_1 ( type_1 var_1 ) { super . method_1 ( var_1 ) ; **var_2 . method_2** ( var_3 ) ; var_2 . method_3 ( var_4 ) ; var_5 . setEnabled ( true ) ; } |
|---|---|
| Expected output | public void method_1 ( type_1 var_1 ) { super . method_1 ( var_1 ) ; var_5 . setEnabled ( true ) ; } |
| Word_base | public void method_1 ( type_1 var_1 ) { var_5 . setEnabled ( true ) ; } |
| span_base | public void method_1 ( type_1 var_1 ) { super . method_1 ( var_1 ) ; var_2 . method_2 ( var_3 ) ; var_2 . method_3 ( var_4 ) ; var_5 . setEnabled ( true ) ; } |
| Text_base | public void method_1 ( type_1 var_1 ) { super . method_1 ( var_1 ) ; var_5 . setEnabled ( true ) ; } |

score based on statements is 0.857, representing slight improvements of 0.003% and 0.037% compared to the other two methods. An analysis of this outcome reveals that all three Seq2edit-based methods, including SLM, effectively retain tokens from the input code by applying "KEEP" labels. Additionally, the SLM method removes redundant statements from the *corpus*, while the word-based and span-based methods are more inclined to retain these redundant statements. A comparison of the outputs for a test sample among the three methods is presented in Table 6. This observation supports the conclusion that the SLM method's annotation strategy, combined with the selective removal of redundant statements, contributes to improved prediction performance and BLEU scores in the CodeBERT model.

Table 6 shows that when predicting a segment of erroneous input code, the red-colored portions highlight the differences between the input code and the expected output code. The test results reveal that only the method proposed in this article produces the correct correction result. The word-based approach results in a missing portion "*super. method_1 ( var_1 )*" in the predicted output, and the span-based method fails to remove the redundant content "*var_2 . method_2 ( var_3 ); var_2 . method_3 ( var_4 )*" from the input sample. This further underscores that although the word-based and span-based methods exhibit favorable BLEU scores, their lower accuracy is a significant contributing factor.

Thus, based on the results of these two sets of experiments, it is demonstrated that the SLM method proposed in this article can automatically annotate the training *corpus*, aiding the error correction model's more effective training. As a result, during prediction, the SLM method achieves optimal error correction results while also considering BLEU values. Compared to existing word-based and span-based methods, the SLM method considers both sentence alignment and word alignment to generate the edit sequence during code alignment. This approach enables the model to predict missing content in a contextually explicit manner accurately.

In summary, the SLM method offers superior performance in code alignment and annotation, effectively enhancing error correction models and achieving optimal error correction results while balancing BLEU values.

## CONCLUSION

The automatic annotation effects of word-based and span-based annotation methods based on seq2edit are not satisfactory. In this article, we propose an annotation method

called SLM based on seq2edit code statement alignment. First, the code is divided into multiple statements according to certain rules, and a cost coefficient is introduced to address the issue that the algorithm cannot handle word-order transposition errors. This improves the Levenshtein algorithm to align code statements, overcoming the problem of the traditional Levenshtein algorithm where tokens belonging to a single code statement are dispersed and aligned to different statements. Second, a beam search algorithm is introduced to search for the optimal matching sentence pairs. By retaining the top K most similar sentence pairs in parallel during each search iteration, the algorithm ensures the highest probability of selecting the optimal alignment sentences. Finally, span-based annotation is applied to the aligned sentences to quickly generate statement annotations, resulting in an output of an edit sequence. Experimental results show that compared to word-based and span-based methods, the annotation results of our method are closer to manual annotation. Moreover, using the annotated *corpus* generated by our method to train and predict with the CodeBERT model, the accuracy and BLEU score reached 15.86% and 0.857, respectively, indicating that our method achieves a good balance in preserving effective tokens in the input code and improving accuracy. In future research, we will improve the prediction model by introducing richer code syntax and semantic representations to decode the missing content of Replace and Insert labels, thereby enhancing the accuracy of generating content for these two types of labels and overall improving the performance of the code correction model.

This article also has some limitations and shortcomings. First, generalization testing is an effective way to verify the effectiveness of our method in correcting different types of code errors. However, due to limitations in resources, the article did not conduct generalization testing on other datasets. Second, Although we employed various methods to reduce randomness during the experimental design process, the high computational cost of model training has thus far precluded us from conducting statistical tests to formally validate that random factors do not influence our results. In future research, we will conduct generalization testing on more diverse code datasets with different types, scales, and domains. Subsequent studies will introduce more sophisticated statistical testing methods, such as repeated experiment design and analysis of variance, to conduct comprehensive statistical analyses of the experimental results, thereby enhancing the practicality and scalability of the model.

### Funding

This work was supported by the Sichuan Science and Technology Program, specifically allocated for the Practice of Collaborative Competition to Promote the Talent Development Model in Education, under grant number 2024NSFSC0499. The funders had no role in study design, data collection and analysis, decision to publish, or preparation of the manuscript.

## Grant Disclosures

The following grant information was disclosed by the authors:
Sichuan Science and Technology Program.
Practice of Collaborative Competition to Promote Talent Development Model in Education: 2024NSFSC0499.

## Competing Interests

The authors declare that they have no competing interests.

## Author Contributions

- Jian Wang conceived and designed the experiments, performed the experiments, analyzed the data, prepared figures and/or tables, and approved the final draft.
- Tao Lin conceived and designed the experiments, authored or reviewed drafts of the article, and approved the final draft.
- Rongsen Zhao performed the experiments, performed the computation work, prepared figures and/or tables, and approved the final draft.
- Huiling Zhao analyzed the data, performed the computation work, authored or reviewed drafts of the article, and approved the final draft.

## Data Availability

The data and code are available at GitHub:

- https://github.com/Ling0924/codebert_gec.

- Huiling Zhao. (2025). Ling0924/codebert_gec: v1.0.0 (v1.0.0). Zenodo. https://doi.org/10.5281/zenodo.15741384

The third-party data is available at GitHub: https://github.com/microsoft/CodeXGLUE.

## Supplemental Information

Supplemental information for this article can be found online at http://dx.doi.org/10.7717/peerj-cs.3024#supplemental-information.

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
