# Peer review of "An enhanced approach for automatic annotation of error codes based on Seq2edit"

_PeerJ Computer Science, doi:10.7717/peerj-cs.3024_

## Round 0.1 · original submission · Major Revisions

Following a thorough evaluation of the feedback provided by the reviewers, we have decided that the manuscript requires major revisions before it can be reconsidered for publication. Below, we summarize the key points raised by the reviewers and outline the necessary steps to address these concerns.
* * *
Key Points and Recommendations
Strengths
• The study is well-motivated, with a clear introduction and relevant references.
• The data, experiments, and results are valid and reasonably well explained, with appropriate conclusions.
• The examples in Tables 1–4 are illustrative and enhance understanding.
• The code repository is publicly accessible, although lacking English documentation.
Major Areas for Improvement
1. Clarity of Methodology:
o Algorithm 2 is unclear in its purpose and integration with Algorithms 1 and 3. Provide a detailed explanation of its inputs, outputs, and role in the overall process.
o Clarify the usage and purpose of key elements such as flag and Levtrans in Algorithm 2.
o The relationship between Equation (1) and the algorithms is not adequately explained.
o Ensure consistent and intuitive notations throughout (e.g., use distinct symbols for words, statements, and codes).
2. Experimental Design:
o The rationale for choosing certain parameters, such as K = 5, should be explained, and if optimization experiments were conducted, include those results.
o Explain the impact of dataset properties (e.g., code length, grammar) on the method’s performance.
o Provide details on data preprocessing, including how noise, outliers, and duplicates were handled in both the CodeXGLUE dataset and the manually annotated pairs.
o Validate the findings with statistical tests to demonstrate the significance of the observed improvements in metrics like ACC and BLEU scores.
3. Repository and Code Documentation:
o The code repository link provided is invalid; ensure the correct link is included.
o Add clear English documentation to the repository, detailing the implementation steps and expected outputs.
4. Results and Evaluation:
o Add statistical tests to validate the significance of the improvements.
o Include more detailed captions for tables and figures to improve clarity.
o Address the absence of edit span counts in Table 5, which is referenced in the text.
5. Language and Writing Style:
o While the writing is generally clear, some sections (e.g., lines 221–229) contain overly formal and inconsistent language, possibly generated using AI tools. Revise these sections to align with the paper's overall tone.
o Revise the abstract to include specific results and clearly emphasize the novelty of the proposed method.
6. Novelty and Contributions:
o Explicitly highlight the novelty and main contributions of the study, particularly how the improved Levenshtein alignment and beam search enhance the annotation process.
7. Generalization and Overfitting:
o Discuss the generalization of the method to other datasets and how overfitting was avoided during model training.
* * *
Minor Comments
• Fix citation formatting (e.g., add parentheses).
• Provide a brief explanation of “overlap” in line 184.
• Clarify and standardize terms and notations in the text and algorithms.
• Clearly separate the two experiments mentioned, ensuring the distinctions are maintained in the evaluation sections.
• Include examples, similar to those in Tables 1–4, to illustrate the outputs of all algorithms.
* * *
Required Actions
1. Point-by-Point Response Letter:
o Address all reviewer comments comprehensively and provide explanations for the changes made.
2. Revised Manuscript:
o Clearly highlight all changes in the revised version for ease of re-evaluation.
o Ensure adherence to the journal's formatting and referencing guidelines.
3. Updated Repository:
o Include the correct link to the repository and ensure that it is well-documented in English.
* * *
Your work on improving sequence alignment methods for automatic code correction is valuable and promising. By addressing the reviewers’ feedback and strengthening the manuscript, we believe this study can make a significant contribution to the field. We look forward to receiving your revised submission.

·

Basic reporting

Strong points:
==============

- The paper is well motivated, the introduction and references are good.
In particular, the examples in Tables 1-4 are very illustrative and helpful
- Data, experiments and results are valid, support the conclusions, and are reasonably well explained.
- Also half of the methodology is well explained (except everything related to Algorithm 2, basically)
- Good English, easy to read, well structured
- Code repo provided online, though explanations are not in English

Comments:
==============

7. The English is unequal in some places, to the level that some generic paragraphs look edited with the help of AI tools, for example lines 221-229 which abound in formalisms
(e.g. "renders it inherently adept for tasks that fall within the purview of programming"
seems out of place compared to the normal-scientific-not-Shakespearean style of the rest of the paper).

Experimental design

The paper presents an improved Seq2edit method of sequence-to-sequence alignment for automatic code correction, with provides word-level alignment and handling of word ordering,
which produces fewer (i.e. more compact) edit changes for the same pair of source and target code snippets.

The results are evaluated in two ways, (i) directly and (ii) through training of a AI model CodeBERT on the
old and enhanced datasets and comparing the prediction results.

There seem to be three levels of matching involved, though this not completely clear from the paper:

- word-based overlap through name similarity/overlap (eq. (2))
- statement-to-statement edit distance, through a modified Levenshtein distance which takes into account order (this is the main contribution)
- a beam search used to find the overall best bipartite alignment among the possible alignments
(expand the search space by next K best candidates, then reducing to the best K, iterate).

Major Comments:
==============

1. The proposed method is only in part well explained, as follows:

Algorithms 1 and 3 are reasonably clear
(Alg1:collect first K best candidates for every statement;
Alg 3: beam search, increase the search space by next K best candidates, then reduce to the best K,
at the end take the best solution)

The problem is with Algorithm 2, which is not clear at all how it fits into the story.
All I understand is that it's some sorted() version of Levenshtein. But:

- Where exactly is it called within Algorithm 1 or 3? Even the outputs `flag` and `Lev_trans` are never used anywhere else

- Are the inputs words, statements or groups of statements (codes)? Until now statements were denoted as X_i and Y_j, and codes as X and Y (see line 158), but the Require of Algorithm 2 says "basic sentences X and Y", which is confusing

- What means "a program error problem" which `flag` indicates? I presume it means it is not just a ordering problem

- In this case, something seems wrong with the algorithm and flag. `flag` = True as soon as we find the ending of X and Y identical up to reordering, even for k=1.
Are you sure it is not != in line 5, or the `flag` values are set correctly, or something else?
As written now, if X and Y are completely different except the last element which is the same,
then `flag` is set to True for k=1 and stays True forever. I don't understand the reason for this.

Later Edit: I also looked in the presumable code found in `tag_seq.py`, `align()` method, but there is no `flag` there

2. Where is equation (1) used in the Algorithms 1,2,3 is also unclear

3. Table 5: edit span count not included, but mentioned in the text in several places

4. The repository " https://github.com/SundayZhao/newRepo"; indicated in the paper submission is not valid. I presume it is https://github.com/SundayZhao/codebert_gec/, but there are no English explanations there, and I didn't see the full name of one of the authors anywhere.

5. It the K in Algorithm 1 (lines 163-167, 5 in Algorithm 1) the same as K in Algorithm 3? If not, it should be made clear.

6. The word-based overlap in lines 183 - 186 is too briefly explained and too quickly glanced over.

7. The English is unequal in some places, to the level that some generic paragraphs look edited with the help of AI tools
for example lines 221-229 which abound in formalisms
(e.g. "renders it inherently adept for tasks that fall within the purview of programming"
seems out of place compared to the normal-scientific-not-Shakespearean style of the rest of the paper).

Minor Comments
==============
- Citations are rendered badly in the pdf, without ()
- line 184: explain shortly what the "overlap" means
- line 191: notations Q don't match the notations U in Algorithm 3
- Algorithm 1: mentions hardcoded value 5, but 5 is just a particular value of K, should be more general
- line 251: "In the experiment...", but above there are mentioned two experiments.
The two experiments should be more clearly separated in all the subsequent evaluation sections,
make it clear when it is the first experiment, and when it is the second experiment.


Suggestions
============
To improve clarity, I would suggest:

- Stick to a common notation: X, Y = codes, X_i, X_j are statements of codes, use some other notation for words of statements if needed (lower case x_i, y_j, or use superscripts)
- Maybe add examples similar to Tables 1-4 to illustrate the outputs of the algorithms
- Explain more clearly the word the part 183 - 186. Make it clear that there are somehow three levels of matching:
(i) word-to word through this overlap cost,
(ii) statement-to-statement edit distance,
(iii) code-to-code by searching for the best statement-to-statement alignment with Algorithm 3 (I presume)

Validity of the findings

- Data, experiments and results are valid, support the conclusions, and are reasonably well explained.
- Conclusions are reasonable.

Reviewer 2 ·

Basic reporting

The manuscript is in clear, professional, and unambiguous English. The introduction strongly motivates the study by establishing the necessity of better annotation methods for automatic error correction in code.
1.The abstract is too general and doesn't specify any results of the proposed methodology.
2. The "cost function," i.e., the function used to align statements (lines 164-186), is not explained clearly. The concept of cost is mentioned, but there’s no intuitive explanation as to how and why it aligns tokens.
3. Also, include how related work describes the course of developing the SLM method.

Experimental design

Three approaches are compared in the experimental design: word-based, span-based, and SLM. 1. Authors kindly explain if the training set's properties such as code length, language grammar, etc. have an impact on SLM's performance.
2. The authors use a publicly available dataset (CodeXGLUE Small) along with 350 manually annotated code pairs. How noisy data was handled and Mention if any outliers, duplicate samples, or irrelevant data were removed.
3. Provide a rationale for selecting K = 5. If experiments were conducted to determine the optimal K value, report those findings.

Validity of the findings

1. The proposed Sentence Label Method (SLM) is novel in its use of improved Levenshtein alignment combined with beam search to enhance the annotation of code errors. The paper lacks a clear statement of novelty. While the approach is novel, it is not explicitly emphasized as a contribution.
2. Add a statistical test to show the significance of the improvements in ACC and BLEU scores.
3. The paper does not mention if the model was validated on a separate dataset or how overfitting was avoided.
4. Quantify the key results in the conclusion.

Additional comments

Add more details on data preprocessing, especially regarding how errors and inconsistencies in the source code were handled. The paper focuses on the CodeXGLUE dataset and 350 manually annotated code samples, but it is unclear if the results generalize to other datasets.Tables and figures lack descriptive captions.

Cite this review as

---

## Round 0.2 · Major Revisions

Based on the reviewer's report, please pay attention to the major changes they have asked you to make.

·

Basic reporting

Clarity and organization has improved in the revision. Introduction, background & motivation are good (not changed from initial version).

One remark:

- Table 5 is hard to understand. I understand the idea of showing how the algorithms are related, and the text explains it, but Table 5 is not clear in its current form. Perhaps use a image? Or a top-level pseudocode description of the whole process?

Experimental design

Algorithm 2 is better explained now in the revised version, which was my main objection in the initial revision. The URL for Github is working now.

Still 4 remarks:

1. The relationship betwen eq (1) and Algorithm 2 is still not clear:
- Algorithm 2 is labeled as "The adjust_Levenshtein algorithm for word order permutation"
- in eq (1) the result is named "adjust_Levenshtein(Xi,Yj)" and it's a product of two things.

Which one is the "adjust_Levenshtein", the Algorithm 2 or eq (1)? They cannot be the same thing. Is eq(1) used somewhere inside Algorithm 2, or is Algorithm 2 computing the first factor in eq(1)? It is not clear. Perhaps it's only a naming issue.

2. I don't understand the logic of Algorithm 2. Flag is initialized as False, it is set to True when a word swap is found in line 5 (sorted() == sorted()), but on the next iterations, it is never reset, even if a difference appears and the if() in line 5 evaluates as false?
As it is described now, if the two statements are completely different and don't have any common word at all, the if() in line 5 never evaluates as true, and Lev_trans remains 0.

3. On data preprocessing, it is mentioned "First, we used the data cleaning tool Numpy to remove noise and irrelevant information from the code". Please be more specific. Numpy is a fundamental Python package for scientific calculations, it's not a data cleaning tool.

4. Line 205: "If the overlap is identical, the Costname is set to 1". What does "overlap is identical" mean? 100% overlap? Then the two sentences are identical up to swaps, since all elements are shared between them?

Validity of the findings

Nothing to change to my initial review:

- Data, experiments and results are valid, support the conclusions, and are reasonably well explained.
- Conclusions are reasonable.

Additional comments

The citations in my review PDF are still looking bad, e.g. lines 302-303, and do not look like the screenshots the authors put in the response. Perhaps the PDF build is producing bad results on the journal side?

---

## Round 0.3 · accepted · Accept

The authors have addressed all of the reviewers' comments. Congrats.

·

Basic reporting

My previous issue was addressed, the unclear table is replaced with a much more clearer figure.

The citations' appearance were fixed.

Experimental design

All issues were addressed:

- The relationship betwen eq (1) and Algorithm 2 is explained
- Algorithm 2 corrected
- Numpy details given
- "overlap is identical" explained

Validity of the findings

No change to my initial review.